# Single-cell mutation identification via phylogenetic inference

Jochen Singer [1,2], Jack Kuipers[1,2], Katharina Jahn[1,2] & Niko Beerenwinkel [1,2]

Reconstructing the evolution of tumors is a key aspect towards the identification of appropriate cancer therapies. The task is challenging because tumors evolve as heterogeneous cell populations. Single-cell sequencing holds the promise of resolving the heterogeneity of tumors; however, it has its own challenges including elevated error rates, allelic drop-out, and uneven coverage. Here, we develop a new approach to mutation detection in individual tumor cells by leveraging the evolutionary relationship among cells. Our method, called SCIΦ, jointly calls mutations in individual cells and estimates the tumor phylogeny among these cells. Employing a Markov Chain Monte Carlo scheme enables us to reliably call mutations in each single cell even in experiments with high drop-out rates and missing data. We show that SCIΦ outperforms existing methods on simulated data and applied it to different real-world datasets, namely a whole exome breast cancer as well as a panel acute lymphoblastic leukemia dataset.

[1] Department of Biosystems Science and Engineering, ETH Zurich, Mattenstrasse 26, 4058 Basel, Switzerland. [2] SIB Swiss Institute of Bioinformatics, 1015 Lausanne, Switzerland. These authors contributed equally: Jochen Singer, Jack Kuipers. Correspondence and requests for materials should be addressed to N.B. (email: niko.beerenwinkel@bsse.ethz.ch)

Due to recent technological advances it is now possible to sequence the genome of individual cells[1]. This allows, for the first time, to directly study genetic cell-to-cell variability and gives unprecedented insights into somatic cell evolution in development and disease.

Having single-cell resolution is especially useful for the analysis of intra-tumor heterogeneity[2]. This is due to the central role that mutational heterogeneity and subclonal tumor composition play in the failure of targeted cancer therapies, where resistant subclones can initiate tumor recurrence[3,4]. Presently, genetic analyses of tumors are mostly based on sequencing bulk samples which only provides admixed variant allele frequency profiles of many thousands to millions of cells. These aggregate measurements are, however, only of limited use for the inference of subclonal genotypes and their phylogenetic relationships[5,6]. The two main issues are that mutational signals of small subclones cannot be distinguished from noise and that the deconvolution of the aggregate measurements into clones is, in general, an under-determined problem.

In contrast, single-cell sequencing data provides direct measurements of cellular genotypes, thus bypassing the deconvolution problem of bulk measurements. However, this advantage comes at the cost of elevated noise due to the limited amount of DNA material present in a cell and the extensive DNA amplification required prior to sequencing[7]. The most common approach for this initial amplification of single-cell DNA is multiple displacement amplification (MDA)[8]. While this process is very efficient at amplifying the overall DNA material, high rates of allelic drop-out, i.e., the random non-amplification of one allele of a heterozygous genotype site, are observed. Starting with the DNA of a single cell, all evidence of a heterozygous genotype mutation is lost when the mutated allele drops out, which happens at a rate of about 10–20%. Also, false positive artifacts can arise in the MDA amplification when random errors introduced early in the process end up with high frequencies due to allelic amplification biases. Further challenges arise from uneven amplification across the genome which results in non-uniform coverage that will leave some sites with insufficient coverage depth for reliable base calling.

These technical issues result in single-cell-specific noise profiles for which regular variant callers developed for next-generation sequencing data, such as the Genome Analysis Toolkit (GATK) HaplotypeCaller[9] or SAMtools[10], are ill-suited. Two single-cell-specific mutation callers, namely Monovar[11] and SCcaller[12], have therefore been recently developed. Both methods take raw sequencing data (BAM files) and output the inferred genotypes of the cells. Monovar specifically addresses the problem of low and uneven coverage in mutation calling by pooling sequencing information across cells, while assuming that no dependencies exist across sites. In contrast, SCcaller detects variants independently for each cell and accounts for local allelic amplification biases. However, the identification of such biases is based on germline single-nucleotide polymorphisms (SNPs), which might not be available, for example, for panel sequencing data. Further, it cannot recover mutations from drop-out events or loss of heterozygosity.

Here, we present SCIΦ, a new single-cell-specific variant caller that combines single-cell genotyping with reconstruction of the cell lineage tree. SCIΦ leverages the fact that the somatic cells of an organism are related via a phylogenetic tree where mutations are propagated along tree branches. SCIΦ can reliably identify single-nucleotide variants (SNVs) in single cells with very low or even no variant allele support and is robust to copy number changes. We show that SCIΦ outperforms Monovar, the only other tool able to transfers information between cells, on simulated and real data.

## Results

**SCIΦ algorithm.** We developed SCIΦ, a probabilistic method for single-cell mutation calling that involves jointly inferring the underlying phylogenetic structure of the cell populations. From the sequencing reads, our inference scheme first identifies candidate loci based on the posterior probability of observing one or more mutated cells at the specific locus. These loci are then used to learn a cell lineage tree employing a Markov Chain Monte Carlo (MCMC) approach. Based on the MCMC posterior sampling, mutations are assigned to cells in a final step. An overview of our method is given in Fig. 1 and details of our approach can be found in Methods section.

**Analysis overview.** In order to investigate the performance of SCIΦ, we conducted several experiments on simulated data and additionally on several real datasets. We compared SCIΦ to Monovar[11], the only previously published single-cell mutation caller sharing information across cells. We start by analyzing the results of the simulated data.

**Benchmarks for simulated data.** We first investigated how the performance depends on the number of cells sequenced in the experiment. SCIΦ is more sensitive in calling mutations than Monovar while showing comparable precision in all settings analyzed (Fig. 2). Therefore, SCIΦ outperforms Monovar with respect to the F1 measure, which is the harmonic mean of precision and recall. The reason for this is twofold: first, due to the tree inference, SCIΦ can assign a mutation to a particular

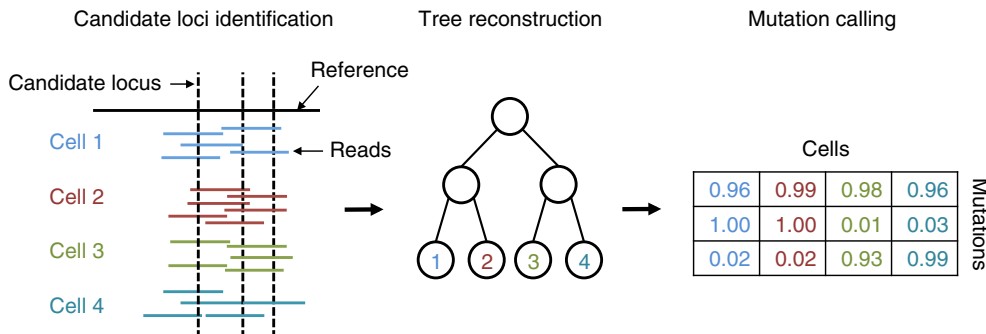

**Fig. 1** Schematic overview of SCIΦ. First, candidate loci are identified. These loci are then used to infer the underlying phylogenetic tree and the parameters of the model. In a last step the mutation to cell assignment is sampled from the posterior distribution

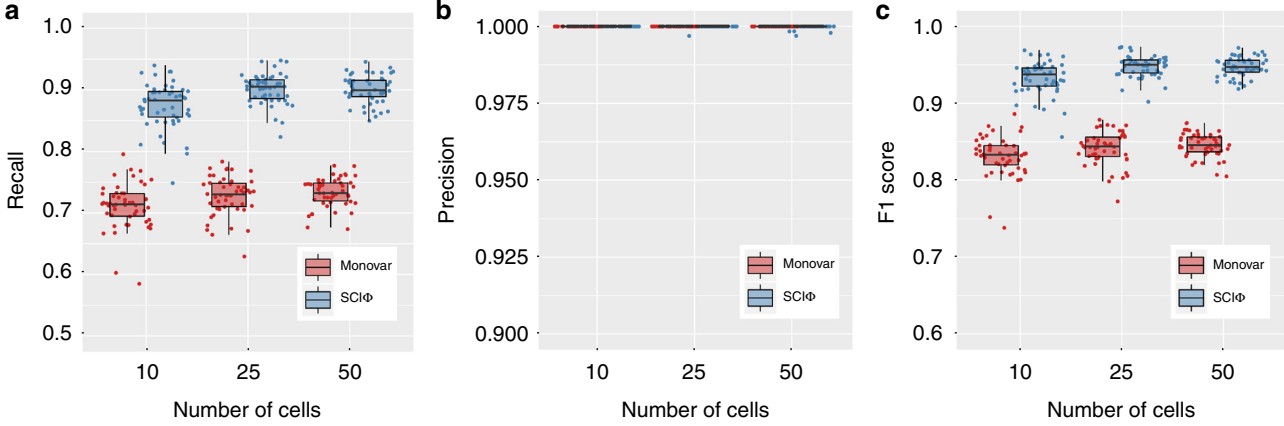

**Fig. 2** Performance of SCIΦ and Monovar on simulated data with different number of cells. Performance measured as recall (**a**), precision (**b**), and F1 score (**c**)

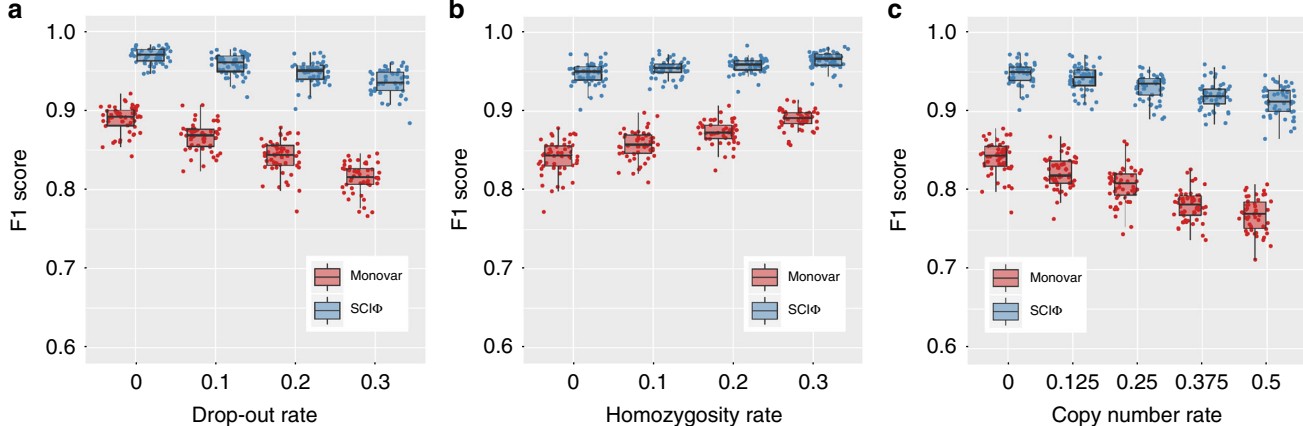

**Fig. 3** Summary statistics of the F1 performance of SCIΦ and Monovar on simulated data. F1 performance depending on different levels of drop-out events (**a**), homozygosity (**b**), and copy number rates (**c**)

cell with very low or even missing variant support at a specific locus. Second, making use of a beta-binomial model to represent the nucleotide counts and learning its parameters accurately reflects the underlying process generating nucleotide counts.

Due to the observed large range of drop-out rates, ranging from 10% to more than 40%[6], a second experiment was conducted to explore the dependence of the methods on the drop-out rate of the experiment. Here we concentrated on drop-out rates of 10, 20, and 30%. Since the exact drop-out rate of a dataset is often not known, we used the default values of the callers, namely 20% for Monovar and 10% for SCIΦ (Fig. 3a).

We found SCIΦ to be more robust to increasing drop-out rates in comparison to Monovar (Fig. 3a). In addition to using the phylogenetic tree structure, SCIΦ also learns the drop-out rate of the experiment during the MCMC scheme and uses 10% only as a starting condition.

An additional experiment was conducted to investigate the effects of loss of heterozygosity. Monovar as well as SCIΦ perform better with increasing levels of homozygous mutations present in the experiment (Fig. 3b). Monovar particularly benefits from homozygous mutations as these are very unlikely to be classified as wild type. SCIΦ experiences a more modest benefit

from homozygous mutations since it already starts with high performance due to the usage of the phylogenetic tree structure to accurately call mutations.

Because copy number events play a prominent role in tumor evolution, we investigated the performance of Monovar and SCIΦ in the presence of additional wild type alleles (Fig. 3c). Similar to the dependence on the homozygosity rate, SCIΦ shows a fairly stable performance for copy number events affecting up to 50% of the mutated loci and outperforms Monovar for all settings. In addition, the performance of Monovar drops more quickly with increasing rate of copy number events.

Additional experiments were conducted to compare Monovar and SCIΦ. We found that both approaches are more suitable to be used on single-cell data than a bulk sequencing mutation caller (Supplementary Section G) and are robust to changes in prior parameters (Supplementary Section D). Further, we show that SCIΦ achieves high accuracy in the tree reconstruction (Supplementary Section I) and that its performance decreases only moderately in the presence of violations of the infinite-sites assumption (Supplementary Section H). Furthermore, a comparison of Monovar and SCIΦ on sequencing data of an isogenic fibroblast cell line (Supplementary Section J) confirms the abovementioned results for simulated data.

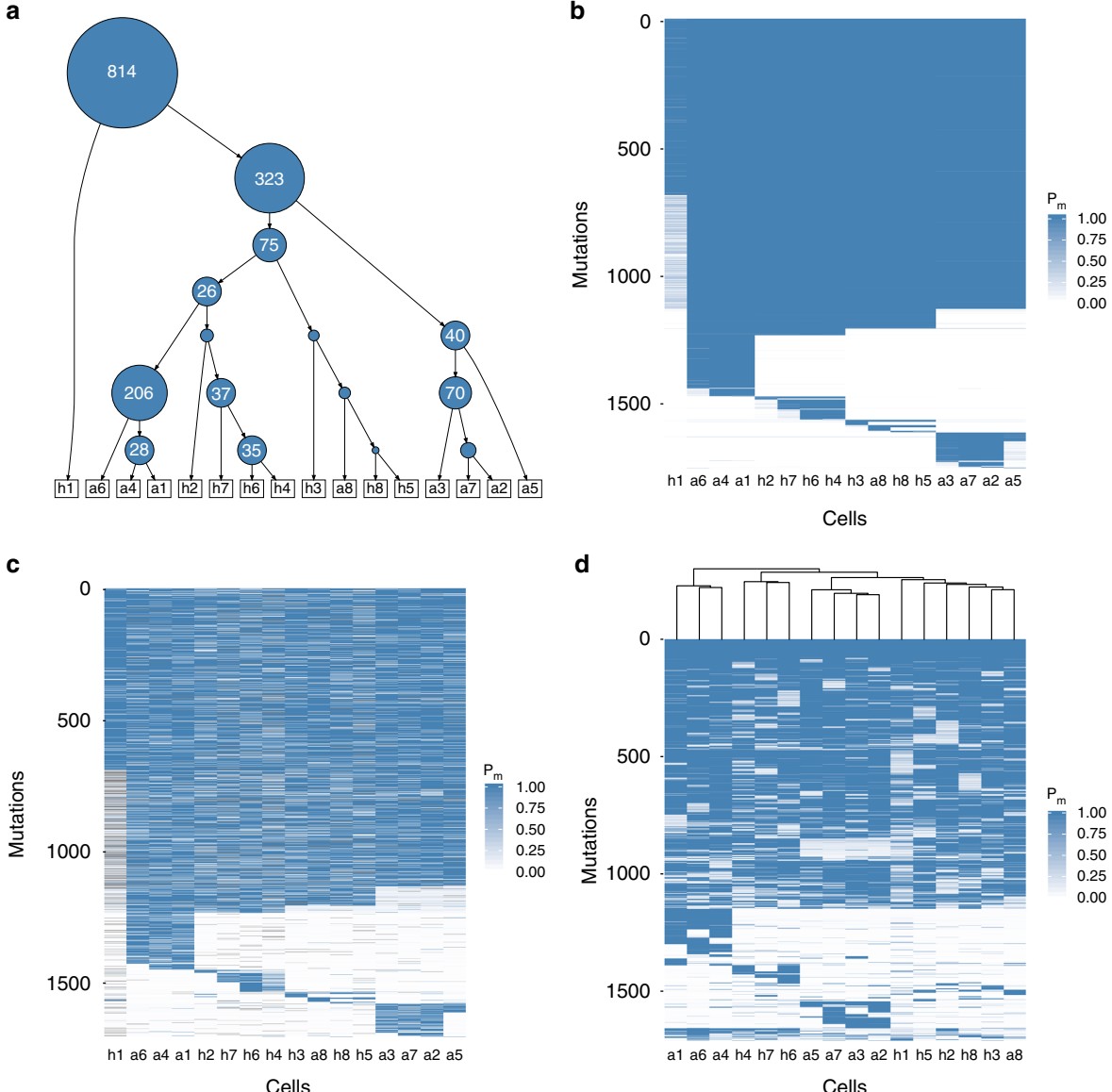

**Fig. 4** Summary of the mutation calls obtained with Monovar and SCIΦ on a breast cancer patient dataset[13] consisting of 16 single tumor cells and a control normal bulk sequencing dataset. **a** Cell lineage tree with average number of mutations per inner node as identified by SCIΦ. The area of a node is proportional to its number of assigned mutations. **b** Posterior probability of SCIΦ mutation calls clustered according to the tree in **a**. **c** Probability of Monovar mutation calls for loci identified as mutated by SCIΦ and clustered according to the tree in **a**. **d** Probability of Monovar mutation calls for loci identified as mutated by SCIΦ and clustered hierarchically

**Application to real data**. We applied SCIΦ to two human tumor sequencing datasets. The first dataset is described in ref. [13], where the authors performed exome sequencing on single cells and bulk samples of a breast cancer patient. Here we identified somatic mutations in 16 single cells using bulk-sequenced normal control dataset to distinguish somatic from germline mutations (see Supplementary Section C for details). This dataset is particularly challenging because cells are aneuploid.

We identified around 50% of the mutations to be shared across all cells and therefore placed them into the root of the inferred phylogenetic tree (Fig. 4a). The average number of mutations assigned to different subclones and their phylogenetic relationship are depicted in Fig. 4a. For example, 323 mutations distinguish cell $h1$ from the other cells and 206 mutations separate the lineage of cells $a1$, $a4$, and $a6$ from the remaining tree. The posterior probabilities of each cell possessing each

mutation show the grouping into subclones (Fig. 4b). Using the tree inferred by SCIΦ to order the mutation calls of Monovar (Fig. 4c) allows a more direct comparison. The assignment of mutations to cells is very homogeneous for the subclones using SCIΦ (Fig. 4b). In contrast, the mutation assignment based on Monovar's inferred probabilities is much more noisy (Fig. 4c).

In order to investigate the impact of using a phylogenetic tree model on the clustering of the cells we performed hierarchical clustering to order the mutation calls from Monovar (Fig. 4d). Hierarchical clustering, which is one of the most widely visualization techniques, leads to a similar subclonal structure compared to SCIΦ (Fig. 4b). However, there are some differences. For example, $h2$ is hierarchically clustered with $h1$, $h3$, $h5$, $h8$, and $a8$, rather than with $h4$, $h6$, and $h7$. The hierarchical clustering does not enforce a phylogenetic tree and weights false negative and false positive signals equally. However, from SCIΦ (Fig. 4a)

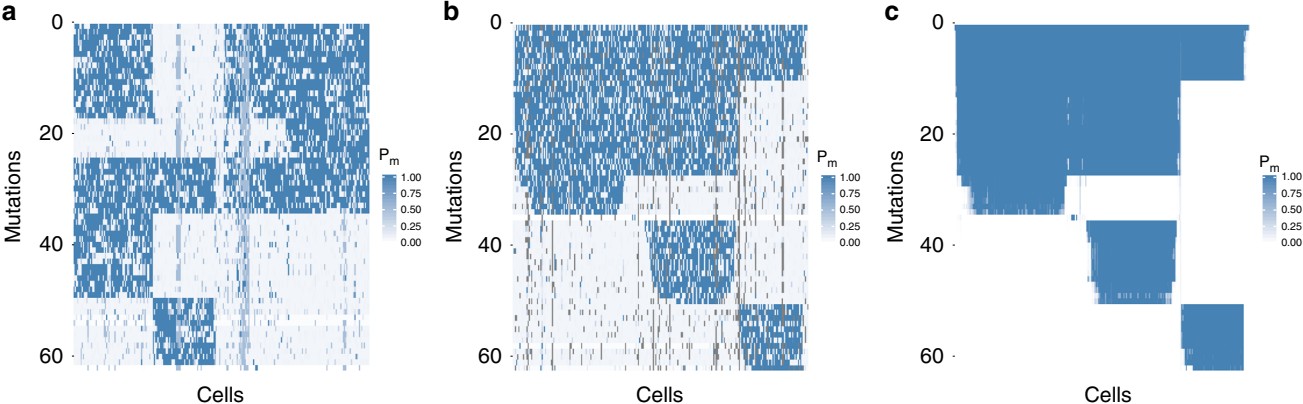

**Fig. 5** Summary of the mutation calls from SCIΦ and Monovar on a dataset consisting of 255 cells from a patient (number 3) with acute lymphoblastic leukemia[14]. **a** Monovar mutation calls for loci identified as mutated by SCIΦ clustered hierarchically. **b** Monovar mutation calls clustered according to the tree inferred by SCIΦ. **c** SCIΦ mutation calls clustered according to its inferred tree

we can see that cell *h2* is only missing mutations which are in common in cells *h4*, *h6*, and *h7*. Therefore, its placement earlier in the tree above those cells is much more evolutionarily plausible.

The second dataset consists of 255 cells from a patient (number 3) with acute lymphoblastic leukemia sequenced using a panel sequencing approach[14]. The results (Fig. 5) highlight similar aspects to those mentioned for the previous breast cancer dataset, especially the much less noisy mutation assignment. It is interesting to observe that SCIΦ not only recovered drop-outs, but also assigned much lower mutation probabilities to likely wild type positions compared to Monovar (Fig. 5).

We obtained results similar to the aforementioned data analyses when analyzing two additional real datasets, namely 19 cells of an isogenic fibroblast cell line with corresponding reference bulk sequencing data and 370 single cells of a high-grade serous ovarian cancer patient (Supplementary Sections J and K). Computational resources are summarized in Supplementary Section L.

## Discussion

Single-cell sequencing allows us to directly study genetic cell-to-cell variability and gives unprecedented insights into somatic cell evolution[15]. This is of particular interest in cancer genomics because tumors show heterogeneous cell compositions often resulting in the failure of targeted cancer therapies. Here, we introduced SCIΦ, the first single-cell mutation caller that simultaneously infers the mutational landscape and the phylogenetic history of a tumor sample. SCIΦ accounts for the elevated noise levels of single-cell data by appropriately approximating the genomic amplification process and the high fraction of drop-out events. In combination with a Markov Chain Monte Carlo phylogenetic tree inference scheme, mutations are reliably assigned to individual cells.

We have compared SCIΦ to Monovar[11] on both simulated and real datasets. For the simulated data, both SCIΦ and Monovar show a precision of almost one, however, SCIΦ shows a substantially higher recall and F1 score. Further, SCIΦ is robust to increasing drop-out, as well as copy number rates. In addition, simulating different MDA amplifications we showed that SCIΦ is not sensitive to the amplification process. For the real datasets, we showed that SCIΦ achieves a much cleaner assignment of mutations to cells within subclones. In particular, SCIΦ recovered mutations from drop-out events using the inferred phylogenetic tree structure of the sample to share information across cells,

whereas Monovar missed these events. Furthermore, the phylogenetic tree inferred by SCIΦ reflects the evolutionary history more accurately than a hierarchical clustering from Monovar results.

Further improvements could be the inclusion of copy number information into the tree reconstruction. However, this comes at the cost of losing the independence between mutation assumption, which is computationally expensive to overcome as groups of mutations would have to be identified.

Mutation calling and lineage tree building are two interdependent tasks and addressing them in a single statistical model provides both improved mutation calls as well as a better estimate of the underlying cell lineage tree, and hence a better understanding of tumor heterogeneity.

## Methods

**Overview.** Our inference scheme starts with an initial identification of possible mutation loci and then performs joint phylogenetic inference and variant calling via posterior sampling (Fig. 1). After introducing the general model for nucleotide frequencies, we describe these steps in more detail. Supplementary Table N provides a summary describing the model parameters.

**Nucleotide frequency model.** We model the nucleotide counts $s$ at a locus with total coverage $c$ using the beta-binomial distribution which is also commonly employed for bulk sequencing mutation detection, e.g.[16,17], as

$$P(s|c, \alpha, \beta) = \binom{c}{s} \frac{B(s+\alpha, c-s+\beta)}{B(\alpha, \beta)}, \quad (1)$$

with parameters $\alpha$ and $\beta$ and where $B$ is the beta function. For better interpretability in our implementation we will employ an alternative parametrization of the beta-binomial distribution with $f = \frac{\alpha}{\alpha+\beta}$ being the frequency of a nucleotide and $\omega = \alpha + \beta$ an overdispersion term determining the shape of the distribution which decreases with increasing variance.

For locus $i$ and cell $j$ with coverage $c_{ij}$, the probability of the observed count (support) $s_{ij}$ for a specific nucleotide in the absence of a mutation is

$$P_{wt}\left(D_{ij}\right) = P\left(s_{ij}|c_{ij}, f_{wt}, \omega_{wt}\right), \quad (2)$$

where $D_{ij} = (s_{ij}, c_{ij})$ and $f_{wt}$ is the expected frequency of the observed nucleotide, which, for example, could have arisen from sequencing error. Large values of $\omega_{wt}$ lead to a binomial distribution representing independent sequencing errors. In the presence of a heterozygous mutation (a mutation affecting one of the two homologous chromosomes), the probability of the counts is

$$P_a\left(D_{ij}\right) = P\left(s_{ij}|c_{ij}, \frac{1}{2} - \frac{2}{3}f_{wt}, \omega_a\right). \quad (3)$$

The underlying allele frequency of $\frac{1}{2}$ is corrected by sequencing errors producing any of the other two bases. Low values of the overdispersion term $\omega_a$ reflect a small

number of initial genomic fragments and any additional feedback in the amplification. SCIΦ generally assumes copy number neutrality, but learning $\omega_a$ allows for additional shifts in the mean variant allele frequency away from $\frac{1}{2}$ due to copy number changes.

**Identification of candidate mutated loci.** Likely mutated loci are identified using the posterior probability of observing at least one mutated cell at a specific locus. The probability of observing no mutation at locus $i$ across all cells is

$$P(K = 0|D_i) = \frac{P(D_i|K = 0)(1 - \lambda)}{P(D_i)}, \qquad (4)$$

where $K$ is a random variable indicating the number of mutated cells and $\lambda$ is the prior probability of a mutation occurring at the locus. The probability of observing the mutation in $k$ cells is

$$P(K = k|D_i) = \frac{P(D_i|K = k)P(K = k)\lambda}{P(D_i)}. \qquad (5)$$

We do not need to compute $P(D_i)$ as it cancels out when computing the likelihood ratio or posterior odds.

The likelihood of the data given that exactly $k$ of the $m$ cells possess the mutation, is given by

$$P(D_i|K = k) = \frac{1}{\binom{m}{k}} \sum_{\{x_1, \dots x_m \in \{0,1\} | \sum x_j = k\}} \prod_{\{j|x_j=1\}} P_a\left(D_{ij}\right) \prod_{\{j|x_j=0\}} P_{wt}\left(D_{ij}\right), \qquad (6)$$

where $x_i$ indicates whether cell $i$ is mutated or not. The term $P(D_i \mid K = k)$ can be computed efficiently using a dynamic programming approach, as in refs. [11,18].

The prior probability of a mutation in a phylogeny affecting $k$ descendant cells is determined by placing mutations uniformly among the edges of the tree (Supplementary Section A) leading to

$$P(K = k) = \frac{\binom{m}{k}^2}{(2k - 1)\binom{2m}{2k}}. \qquad (7)$$

**Allelic drop-out.** Along with the uncertainty in the supporting read counts due to the amplifications in each cell when a mutation is present, an additional artifact is drop-out whereby one allele is not amplified at all. To account for allelic drop-out occurring with probability $\mu$, we introduce the following mixture for the likelihood of the observations for each cell:

$$P_a\left(D_{ij}\right) = \frac{\mu}{2}P\left(s_{ij}|c_{ij}, f_{wt}, \omega_{wt}\right) \\ + \frac{\mu}{2}P\left(c_{ij} - s_{ij}|c_{ij}, f_{wt}, \omega_{wt}\right) \qquad (8) \\ + (1 - \mu)P\left(s_{ij}|c_{ij}, \frac{1}{2} - \frac{2}{3}f_{wt}, \omega_a\right),$$

where the first term describes the loss of the mutant allele, the second the loss of the wild-type allele and the third term describes a heterozygous mutation. The case $\mu = 0$ reduces to Eq. (3).

**Tree likelihood.** Different approaches for single-cell phylogeny reconstruction have been proposed[7], including OncoNEM[19] and SCITE[20]. Our model to infer tumor phylogeny consists of three parts, akin to ref. [20]: the tree structure $T$, the mutation attachments to edges $\sigma$, and the parameters of the model $\theta$ (the parameters $f_{wt}$, $\omega_{wt}$, and $\omega_a$ previously introduced, the drop-out mixture coefficient $\mu$ as well as a homozygosity coefficient which we will introduce later). We represent the phylogeny of a tumor using a genealogical tree. Here the $m$ sampled tumor cells are represented by leaves in a binary tree and the mutations are placed along the edges. There are $(2m - 3)!!$ different tree structures[21], while each of the $n$ mutations can be attached to the $(2m - 1)$ edges leading to $(2m - 3)!!(2m - 1)^n$ possible configurations for the discrete component $(T, \sigma)$ of our model. As a result, it is infeasible to enumerate all solutions. Instead we employ a Markov Chain Monte Carlo approach to search and sample from the tree space.

In order to do so, we employ the likelihood of a specific tree realization with the mutation attachment parameter $\sigma$ and the parameters $\theta$ to be

$$P(D|T, \sigma, \theta) = \prod_{i=1}^{n}\prod_{j=1}^{m} P\left(D_{ij}|T\right)\prod_{i=n+1}^{N}\prod_{j=1}^{m} P_{wt}\left(D_{ij}\right), \qquad (9)$$

where $P(D_{ij} \mid T) = P_a(D_{ij})$ if the cell $j$ is below mutation $i$ (on the path from leaf $j$ to the root) and $P(D_{ij} \mid T) = P_{wt}(D_{ij})$ otherwise. The first set of products describes the loci identified to be likely mutated (section Identification of candidate mutated loci)

which are placed on the tree and used together to infer its phylogenetic structure. The second half represents all loci where no mutation is present which inform the inference of the sequencing error parameters.

We marginalize out the attachment points of the mutations, analogously to ref. [20]. Assuming each mutation is equally likely to attach to any edge in the tree and the attachment probability to be independent between mutations we have $P(\sigma \mid T, \theta) = \frac{1}{(2m-1)^n}$ so that

$$P(D|T, \theta) \propto \sum_{\sigma} \prod_{i=1}^{n}\prod_{j=1}^{m} P\left(D_{ij}|T\right) = \prod_{i=1}^{n}\sum_{\sigma_i}\prod_{j=1}^{m} P\left(D_{ij}|T\right), \qquad (10)$$

For each locus, the sum over $\sigma_i$ can be written explicitly as

$$S_a(D_i|T) = \frac{1}{2m-1}\sum_{\sigma_i}\prod_{j=1}^{m} P\left(D_{ij}|T\right) \\ = \frac{1}{2m-1}\sum_{\sigma_i}\prod_{j=1}^{m}\left[I(\sigma_i \prec j)P_a\left(D_{ij}\right) + I(\sigma_i \nprec j)P_{wt}\left(D_{ij}\right)\right], \qquad (11)$$

where $I$ is the indicator function and $(\sigma_i \prec j)$ indicates that cell $j$ sits below the attachment point $\sigma_i$ of mutation $i$ in the tree $T$. The sum can be computed in $O(m)$ time using the binary tree structure. Employing $T$, we propagate the probability of attaching a mutation to a specific node from the leaves toward the root. This can be implemented using the depth-first search (DFS) algorithm, combining in each node the probabilities from two previously computed subtrees.

Computing Eq. (10) is therefore in $O(mn)$ while the marginalization has the benefit of reducing the search space by a factor of $(2m - 1)^n$. In addition we employ the marginalization to focus on the tree structure of the cell lineage rather than the attachment points of mutations.

Making use of the factorization of the beta-binomial density function into Gamma functions, the term $\prod_{i=n+1}^{N}\prod_{j=1}^{m} P_{wt}\left(D_{ij}\right)$ in Eq. (9) can be computed in time linear in the number of different coverages of the sequencing experiment (Supplementary Section B). Since that number is typically much smaller than $mn$, the overall runtime is dominated by $O(mn)$.

**Accounting for zygosity.** Because tumor cells show chromosomal abnormalities, mutations can be observed as homozygous variants even without drop-out events. In order to also account for loss of heterozygosity, we adapt the scheme introduced in section Tree likelihood. Instead of computing the likelihood of the data when attaching a mutation to a node in the lineage tree in the heterozygous state only, we additionally compute the likelihood when attaching each mutation in the homozygous state, and define the sum

$$S_h(D_i|T) = \frac{1}{m-1}\sum_{\sigma_i}\prod_{j=1}^{m} P\left(D_{ij}|T\right) \\ = \frac{1}{m-1}\sum_{\sigma_i}\prod_{j=1}^{m}\left[I(i \prec j)P_h\left(D_{ij}\right) + I(i \nprec j)P_{wt}\left(D_{ij}\right)\right], \qquad (12)$$

involving the nucleotide model when only alternative alleles are present

$$P_h\left(D_{ij}\right) = P\left(c_{ij} - s_{ij}|c_{ij}, f_{wt}, \omega_{wt}\right). \qquad (13)$$

Note that homozygous mutations are only attached to inner nodes as the probability of observing a drop-out event in a single cell is assumed to be higher than a single homozygous mutation.

Utilizing the tree structure, the sum can again be computed in $O(m)$ time for each mutation on the tree. The overall likelihood (Eq. (10)) for each mutation becomes a weighted sum of the two possibilities leading to

$$P(D|T, \theta) \propto \prod_{i=1}^{n}[(1 - \nu)S_a(D_i|T) + \nu S_h(D_i|T)], \qquad (14)$$

with homozygosity coefficient $\nu$. Thus, we allow certain violations of the infinite-sites assumption[22] by capturing homozygous mutations which are not due to drop-out events.

**Markov Chain Monte Carlo sampling.** Using the tree likelihood, we employ an MCMC scheme to sample from the posterior distribution of mutation assignments as well as tree structures given the data (for simplicity with uniform priors). In order to do so, we propose a new state $(T', \theta')$ from the current state $(T, \theta)$ making use of properly defined moves, described below, such that the chain is ergodic. We change one parameter at a time with transition probability $q(T', \theta' \mid T, \theta)$ and accept the new configuration with probability

$$\min\left\{1, \frac{q(T, \theta|T', \theta')P(T', \theta'|D)}{q(T', \theta'|T, \theta)P(T, \theta|D)}\right\}. \qquad (15)$$

The tree structure can be changed using the prune and reattach move. Here we randomly draw a node from the tree and re-attach it to a random node not

contained in the pruned subtree. This move is reversible, irreducible, and aperiodic. Additionally we include a move which swaps two leaf nodes. For the parameters of the beta-binomial distribution, the drop-out coefficient $\mu$ (and the homozygosity coefficient $\nu$) we perform independent random Gaussian walks. The standard deviations of the steps are adjusted using adaptive MCMC[23] to track an acceptance rate of 50%.

We sample proportional to $P(T, \theta \mid D)$ from the posterior distribution after a burn-in phase. Convergence is achieved after $x$ iterations, with heuristic arguments suggesting $x \propto m^2 \log(m)$[20], and can be verified by computing the correlation between two runs in practice. The overall runtime complexity is $O(x \times \max(mn, c))$ with $c$ being the number of unique coverage values of the experiment. From the sample of trees and parameters we could also conditionally sample the placement of the mutations for the full joint posterior sample. Instead, utilizing the full weights of attaching each mutation to different edges we record the probability of each cell possessing each mutation. Averaging over the MCMC chain provides the posterior genotype matrix and hence our single-cell variant calls.

**Simulation of ground truth datasets.** In order to benchmark the performance of SCIΦ, we simulated tumor evolution by introducing a cell lineage tree and simulated read counts by mimicking the noisy MDA process. For $m$ cells, we created a random binary genealogical cell linage tree with 100 mutations attached to the edges. The placement of the mutations defines which cells possess each mutation. We chose the placement such that each mutation is shared by at least two cells because mutations in only one cell may be false positives from sequencing errors and are filtered out in practice as well as in our benchmark. Further, among all the mutations present in cells a specified fraction $\mu$ was randomly selected as drop-outs, i.e., $\frac{\mu}{2}$ of the mutations became wild type and $\frac{\mu}{2}$ became homozygous alternative genotype.

Then we generated an artificial reference chromosome of 1 million base pairs (bp) and divided it into segments of ~1000 bp for each cell individually. For these segments, we generated a coverage distribution following a negative binomial distribution with a mean of 25 nucleotides and a variance of 50. Additionally, 10% of the segments were assigned 0 coverage to include missing information. The coverage $c$ of specific positions was additionally randomized following a discretized Gaussian distribution with the segment coverage as mean and a standard deviation of 10% of that mean in order to simulate the uneven coverage profiles of real single-cell sequencing experiments.

For simulating nucleotides under the MDA process, we drew them from a Pólya urn model. Because data suggest that the two homologous chromosomes are amplified independently of one another (ref. [24] and Supplementary Figure M), we chose to set the initial number of alleles ($\alpha$ and $\beta$) to 1 for heterozygous genotypes (which would lead to a uniform distribution without errors and drop-out). For homozygous genotypes either $\alpha$ or $\beta$ were set to 1. An allele is then randomly chosen, copied, and returned to the urn together with the copy. With a probability of $5 \times 10^{-7}$ the copy will be mutated and an allele different from the original one is returned, corresponding to the error rate of the MDA polymerase ($10^{-6}$–$10^{-7}$ [25]). This process is repeated $c$ times and the copies are retained. In order to simulate copy number events, we change the number of initial copies of the wild type allele for a specific locus. We set the probability of $x$ extra copies to be $\frac{1}{2^x}$, since each additional copy is less likely. This strategy assumes all copy number changes happened prior to mutation events. In reality this is not true, however, the strategy provides lower bounds on the performance measures because the variant allele frequency decreases with increasing copy number. Finally, with probability of $10^{-3}$, a nucleotide is mutated to account for sequencing errors, and the resulting simulated data was embedded into a multi-pileup file.

Since the MDA amplification process is not fully understood and different models of dependence between homologous chromosomes have been proposed[24] we performed additional simulations (Supplementary Section E) for the model of dependence reported, for example, in ref. [15]. In addition, since different amplification techniques, such as MALBAC[26] or pure PCR based methods, are also employed, we simulated different amplification scenarios (Supplementary Section F). Both experiments were in line with the previously reported results.

The simulation framework was implemented using Snakemake[27] and can be found at https://github.com/cbg-ethz/SCIPhI. Simulations were replicated 50 times. All box plots were generated using ggplot2[28] and the data points overlaid.

**Code availability.** SCIΦ has been implemented in C++ using ref. [29] and is freely available under a GNU General Public License v3.0 license at https://github.com/cbg-ethz/SCIPhI.

## Data availability

The human sequencing datasets utilized in this study were downloaded from the Sequence Read Archive with the accession numbers SRA053195 (for the dataset generated in ref. [13]) and SRP044380 (for the dataset of patient three generated in ref. [14]).

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

## Acknowledgements

We thank David Seifert for constructive discussions and C++ support as well as Franziska Singer for critical feedback. J.S. and J.K. were supported by ERC Synergy Grant 609883 (http://erc.europa.eu/). K.J. was supported by SystemsX.ch RTD Grant 2013/150 (http://www.systemsx.ch/).

## Author contributions

J.S., J.K., and N.B. designed the study. J.S. and J.K. developed the methodology. J.S. implemented the methodology. J.S., J.K., and K.J. performed analyses. All authors drafted the manuscript and approved the final version.

## Additional information

**Competing interests:** The authors declare no competing interests.

