## [Peer Review File · Nature Communications]

Reviewers' comments:

Reviewer #1 (Remarks to the Author):

This paper describes a new variant caller for point mutations in single cell DNA sequence, SCIΦ. The work addresses a pressing need for better computational tools to work with single-cell sequence data, in particular with respect to the relatively underdeveloped areas of single-cell DNA-seq tools. As the paper notes, generic bulk sequence variant callers are poorly suited to the task of calling from single-cell data and there are so far few alternatives designed specifically for single-cell data. The problem is well motivated and paper makes a good case for the idea that a caller able to take account of common ancestry among cells will prove more effective, particularly for the cancer application. The results, in comparison to a competitor tool Monovar, look very promising, and the authors make their code freely available under appropriate terms. There are some points where the work might be strengthened, noted below, as well as some aspects where the software might have been taken farther (e.g., copy number variant calling, as the authors note, or structural variant calling, which they do not). Nonetheless, it appears a valuable and timely contribution to the literature and an advance over the state of the art in the space.

The probabilistic model behind the method is involved but well justified in the details. The mutation model is reasonable, as are models for read depths. The phylogenetic likelihood is likewise well supported by the literature and appropriate for the problem. The MCMC sampling method is also fairly standard and appropriate, with good theoretical grounds for believing it will yield efficient samples on realistic data sets.

The phylogeny model could in principle be more sophisticated, in terms of allowing for variation in edge lengths, mutation rates, and so forth, typical of state-of-the-art Bayesian phylogeny methods, but it is reasonable for getting a working method in the face of relatively limited data. The MCMC algorithm, like the phylogeny model, does not necessarily make use of the most advanced methods coming into use for other phylogenetic problems in this domain (e.g., approximate Bayesian computation or sequential Monte Carlo). However, the empirical data makes a good case that that sort of machinery would be overkill for the current model, even if they might be useful in future work with more complex phylogeny variants.

One apparent exception to the plausibility of the model is the assumption of copy number neutrality, which would appear *prima facie* to be problematic for cancer genomes. One might expect the method to be more effective with a more principled model of copy number variation, as well as to provide more useful data on allelic variants in non-diploid regions. The paper provides a good empirical demonstration, though, on simulated and real data, that the model is actually quite robust to copy number variations and it is reasonably considered as an avenue for future work.

One point that might be worth describing more explicitly is dependence on the single-cell technology used. The authors highlight a specific amplification technology (MDA) and the assumption of an MDA-like amplification step appears to be built into the model, but there are alternative amplification methods (e.g., MALBAC) as well as low-depth amplification-free alternatives for single-cell DNA-seq (e.g., Zahn et al., *Nature Methods*, 14:167, 2017). At a minimum, a discussion is warranted on alternative single-cell sequencing technologies and which of these would be amenable to Sciphi. Even better would be to extend the empirical evaluation to cover simulated and real data from one or more alternative techniques.

The empirical validation is generally appropriate and convincing. There is a fair comparison to Monovar, which appears to be the only clear competitor in the space so far. The real data sets are well chosen

and appropriate for the problem, covering two cancer types and tumors known to include likely confounding effects such as aneuploidy. The paper uses a reasonably designed protocol for generating synthetic data, accounting for plausible sources of error in variant calling and phylogeny inference, with a couple of possible exceptions. It might be worth exploring explicitly how robust the method is to hypermutable sites, selection-driven reversion, or other likely sources of homoplasy, given that homoplasy has been widely observed in tumor genomes and would seem a likely confounding factor for the phylogenetics. I am also curious if misalignment of reads due to repetitive DNA would be a source of error in this analysis, something that appears not to be considered in the simulations. Semi-simulated data, using a real human chromosome as a starting point, might give a better control than a truly random initial genome.

The results do appear to make a convincing case on simulated and real data that the tree model of SCIΦ leads to significantly improved calls relative to the more generic model of Monovar. The study of precision and recall shows clear improvement over Monovar by what appears to be a fair test across a range of plausible simulation parameters. The results on real data likewise appear visually to make a compelling case for cleaner and more accurate calls with SCIΦ. One minor point is that it would be useful to see some quantification of the claim that Monovar's results are more noisy than SCIΦ's; while it appears obvious from Figures 4 and 5, visually obvious results are not always correct. One other minor quibble is that, while the paper reasonably compares the SCIΦ results to those of a tree derived directly from Monovar, hierarchical clustering is not necessarily the soundest way to make a tree from the Monovar calls. The use of hierarchical clustering ought to be better defended, or alternatively replaced with a more proper generic phylogeny tool (e.g., Paup* or MrBayes).

The paper is well written, organized, and clear. Background and references are appropriate, although they could perhaps better cite work to date on phylogenetics from single-cell sequence data specifically. There was a recent review from Zafar et al. (Current Opinion in Systems Biology, 2017) that might be helpful there. I noted just one minor error: ``substantial'' should be ``substantially'' on p. 11 line 275.

Reviewer #2 (Remarks to the Author):

The authors propose a brand new computational method to infer mutations for single cell sequencing data based on jointly inferring the phylogenetic tree of cells. The idea is excellent and is expected to provide new methodological insights into the detection problem of somatic mutations. The implementation of the method is also elegant, reaching superior performance compared to Monovar. However, before acceptance for publication, some concerns should be clarified.

1. The basic idea of the proposed method is jointly inferring the cellular phylogenetic tree and the mutations at almost all the loci. In general, these two tightly-linked questions are solved separately. Employing the linkage of these two questions to refine the inference of each other is creative. However, it seems that the authors did not sufficiently evaluate the performance of the joint inference. Particularly, the authors only evaluated the accuracy of the new method for mutation detection, without validating the reliability of the inferred phylogenetic tree(s). If the phylogenetic tree is not significantly improved, the refined mutation detection needs a more persuasive explanation.
2. According to my understanding, the joint inference is conducted separately for each locus. However, the true phylogenetic tree of a specific cell population seems to be unique, and more importantly the contribution of each locus to the construction of the cell phylogenetic tree varies greatly. My concern is how consistent the inferred trees were for different loci. If the consistency is under expectation, how

did the trees guide the inference of mutations? The authors should show example trees inferred from different loci so that the readers can get intuitive impressions of the performance of the proposed method.

3. As the authors mentioned, the joint inference results in huge computational complexity. The authors should show the running time and memory usage for different datasets and compare to other softwares. Only Monovar is not sufficient to demonstrate the relative performance of the proposed method. Mutect, Strelka and other frequently used tools should also be included.

4. Only two real datasets were included to evaluate the tool performance, and the cell numbers were only 16 and 255. Datasets with more cells should be evaluated.

5. Another concern is the robustness of the inference. Due to the computational complexity, the authors used MCMC to complete the inference. The authors should demonstrate the robustness of the inferring results by adding replications. Furthermore, quality control is a critical step before mutation detection. The authors should also show whether the inference is robust to various quality control criteria.

Reviewer #3 (Remarks to the Author):

The manuscript by Singer et al describes SCIΦ, a new statistical approach for detecting variants from single cell DNA sequence data. This is a very important topic as currently most single cell DNA analysis projects use algorithms that are designed for bulk samples and have poor performance. Their new statistical framework exploits the power gained by inferring the underlying phylogenetic tree of a collection of multiple cells from the same sample to mitigate errors in single cell data, including false positive errors, coverage unevenness, allelic dropout and loss of heterozygosity. Using the controlled experiment of a simulated single cell dataset, they demonstrate that SCIΦ has greatly improved sensitivity and specificity over the currently existing single cell mutation identification methods and the widely used algorithms such as Samtools and GATK UnifiedGenotyper. They then apply their new method to real single cell DNA sequencing datasets from two human tumor samples (a triple-negative breast cancer and an acute lymphoblastic leukemia cancer), detect somatic mutations and infer the clonal structures of the two tumors. The manuscript is very well written and interesting, although this reviewer has the following concerns:

1. The authors compared SCIΦ with Monovar, another single cell SNV detection method and demonstrated that SCIΦ outperforms Monovar. The performance improvement of Monovar mostly comes from the error model used by the algorithm. The authors of Monovar cultured an isogenic cell line, then sequenced multiple individual cells from the cell line. They then compared single cell data against bulking sequencing results of the same cell line. Assuming mutational differences are all errors, they built the errors model of single cell DNA sequencing and filtered these errors from real single cell datasets. Monovar also counts only mutations that are shared by at least two cells. So, my question here is, where does the performance improvement of SCIΦ come from?

2. Is SCIΦ able to detect small insertions and deletions using single cell DNA data? Could the authors evaluate the performance of SCIΦ for detecting INDELS?

3. The authors benchmarked SCIΦ using a simulated single cell DNA sequencing dataset. Even though they tried to spike-in random false positive errors, allelic dropouts etc., the error pattern of simulated data is always different from that of real single cell data. Will the authors be able to benchmark SCIΦ using an isogenic cell line data?

4. The authors applied SCIΦ to two real single cell datasets. However not all the mutations from the

two datasets are validated. My understanding is that many mutations, including mutations that are detected in just one cell, from the triple negative breast cancer tumor were validated using an alternative sequencing method. How does SCIΦ perform using validated single cell mutations only?

5. In figure 4a, the authors inferred the clonal structures of the triple negative breast cancer using mutations detected by SCIΦ and classified cells h3, h5, h8 and a8 into one sub-clone. However, the Nature paper by Wang et al. suggests that cells h3, h5 and h8 are hypodiploidy cells while a8 is an aneuploidy cell. These two groups of cells have very different genotypes based on flow sorting results and single cell mutations. How were they grouped together?

6. SCIΦ counts only mutations that are shared by at least two cells. However, when the authors studied the clonal structures of the triple negative breast cancer using mutations identified by SCIΦ, they found that cell h1 is an orphan cell. H1 shares all its mutations with at least one other cells but it is also an orphan. How come?

7. In figure 4d, the authors classified cells from the triple negative breast cancer using Monovar mutations. However, the classification result is very different from the result of the Monovar paper. What has changed?

Reviewer #1 (Remarks to the Author):

This paper describes a new variant caller for point mutations in single cell DNA sequence, SCIΦ. The work addresses a pressing need for better computational tools to work with single-cell sequence data, in particular with respect to the relatively underdeveloped areas of single-cell DNA-seq tools. As the paper notes, generic bulk sequence variant callers are poorly suited to the task of calling from single-cell data and there are so far few alternatives designed specifically for single-cell data. The problem is well motivated and paper makes a good case for the idea that a caller able to take account of common ancestry among cells will prove more effective, particularly for the cancer application. The results, in comparison to a competitor tool Monovar, look very promising, and the authors make their code freely available under appropriate terms. There are some points where the work might be strengthened, noted below, as well as some aspects where the software might have been taken farther (e.g., copy number variant calling, as the authors note, or structural variant calling, which they do not). Nonetheless, it appears a valuable and timely contribution to the literature and an advance over the state of the art in the space.

The probabilistic model behind the method is involved but well justified in the details. The mutation model is reasonable, as are models for read depths. The phylogenetic likelihood is likewise well supported by the literature and appropriate for the problem. The MCMC sampling method is also fairly standard and appropriate, with good theoretical grounds for believing it will yield efficient samples on realistic data sets.

The phylogeny model could in principle be more sophisticated, in terms of allowing for variation in edge lengths, mutation rates, and so forth, typical of state-of-the-art Bayesian phylogeny methods, but it is reasonable for getting a working method in the face of relatively limited data. The MCMC algorithm, like the phylogeny model, does not necessarily make use of the most advanced methods coming into use for other phylogenetic problems in this domain (e.g., approximate Bayesian computation or sequential Monte Carlo). However, the empirical data makes a good case that that sort of machinery would be overkill for the current model, even if they might be useful in future work with more complex phylogeny variants.

1) One apparent exception to the plausibility of the model is the assumption of copy number neutrality, which would appear *prima facie* to be problematic for cancer genomes. One might expect the method to be more effective with a more principled model of copy number variation, as well as to provide more useful data on allelic variants in non-diploid regions. The paper provides a good empirical demonstration, though, on simulated and real data, that the model is actually quite robust to copy number variations and it is reasonably considered as an avenue for future work.

Thank you for this comment! We agree that copy number changes can play an important role in some cancers. As already pointed out by the reviewer, SCIΦ

performs well for a range of copy number changes. The inclusion of an explicit copy number model is not straightforward, most notably because the mutations can no longer be treated independently. Without this assumption the complexity of the problem increases considerably, requiring drastic changes of the underlying approach. For this reason, incorporating an explicit copy number model exceeds the scope of this project, but should be considered in future work.

2) One point that might be worth describing more explicitly is dependence on the single-cell technology used. The authors highlight a specific amplification technology (MDA) and the assumption of an MDA-like amplification step appears to be built into the model, but there are alternative amplification methods (e.g., MALBAC) as well as low-depth amplification-free alternatives for single-cell DNA-seq (e.g., Zahn et al., *Nature Methods*, 14:167, 2017). At a minimum, a discussion is warranted on alternative single-cell sequencing technologies and which of these would be amenable to Sciphi. Even better would be to extend the empirical evaluation to cover simulated and real data from one or more alternative techniques.

We thank the reviewer for their comments. The beta-binomial model is not restricted to model nucleotide distributions from the MDA process, but with different parameter values has been used for several mutation callers (e.g. deepSNV [1], deNovoGear [2]) designed for traditional bulk sequencing data. Therefore, our approach to model the nucleotide distribution also covers other approaches, including MALBAC. In fact, MALBAC uses isothermal amplification (usually MDA) in a first step, before a second round of traditional PCR amplification, such that the amplification method can be seen as a combination of the MDA and PCR amplification approach. In order to emphasize the general usage of the beta-binomial model, we added the sentence “We model the nucleotide counts s at a locus with total coverage c using the beta-binomial distribution that has previously been used for traditional bulk sequencing mutation calling [8, 30] as ...” in line 74. Further, we conducted an additional simulation with the initially present copies of the homologous chromosomes set to 10, 50 and 100 (Supplementary Section “Influence of the amplification process”). In doing so, our simulation becomes closer to the traditional PCR amplification and covers different amplification schemes.

[1] Gerstung, M., Beisel, C., Rechsteiner, M., Wild, P., Schraml, P., Moch, H., & Beerenwinkel, N. (2012). Reliable detection of subclonal single-nucleotide variants in tumour cell populations. *Nature communications*, 3, 811.

[2] Ramu, A., Noordam, M. J., Schwartz, R. S., Wuster, A., Hurles, M. E., Cartwright, R. A., & Conrad, D. F. (2013). DeNovoGear: de novo indel and point mutation discovery and phasing. *Nature methods*, 10(10), 985.

3) The empirical validation is generally appropriate and convincing. There is a fair comparison to Monovar, which appears to be the only clear competitor in the space so far. The real data sets are well chosen and appropriate for the problem, covering two cancer types and tumors known to include likely confounding effects such as aneuploidy. The paper uses a reasonably designed protocol for generating synthetic data, accounting for plausible sources of error in variant calling and phylogeny inference, with a couple of possible exceptions. It might be worth exploring explicitly how robust the method is to hypermutable

sites, selection-driven reversion, or other likely sources of homoplasy, given that homoplasy has been widely observed in tumor genomes and would seem a likely confounding factor for the phylogenetics. I am also curious if misalignment of reads due to repetitive DNA would be a source of error in this analysis, something that appears not to be considered in the simulations. Semi-simulated data, using a real human chromosome as a starting point, might give a better control than a truly random initial genome.

We agree that homoplasy might be a confounding factor and could violate the infinite site assumption underlying SCIΦ. Therefore, in the Supplementary Section “Violation of the infinite sites assumption”, we investigate the performance of SCIΦ in the presence of violations to the infinite sites assumption. In order to do so, we simulated specific mutations that are independently accumulated twice in different branches of the tree and we simulate the loss of a mutation.

Regarding semi-simulated data, in Supplementary Section “Performance comparison using isogenic cell line data” we analyze the performance of Monovar and SCIΦ in terms of recall, precision, and F1 score using an isogenic fibroblast cell line. It is worthwhile noticing that the data set is particularly challenging for SCIΦ as we do not expect a pronounced phylogenetic structure within an isogenic cell line. Therefore, SCIΦ should assign most of the mutations to all cells, which is the case as shown in Supplementary Figure 15. The inferred cell lineage tree has a mostly linear structure with some branching close to the leaves. The linear structure is a result of noise in the data, such that some of the drop-out events are not detected as such. However, SCIΦ assigns the vast majority of mutations to the root of the inferred cell lineage tree and the first three levels already contain 94% of all mutations.

4) The results do appear to make a convincing case on simulated and real data that the tree model of SCIΦ leads to significantly improved calls relative to the more generic model of Monovar. The study of precision and recall shows clear improvement over Monovar by what appears to be a fair test across a range of plausible simulation parameters. The results on real data likewise appear visually to make a compelling case for cleaner and more accurate calls with SCIΦ. One minor point is that it would be useful to see some quantification of the claim that Monovar’s results are more noisy than SCIΦ’s; while it appears obvious from Figures 4 and 5, visually obvious results are not always correct. One other minor quibble is that, while the paper reasonably compares the SCIΦ results to those of a tree derived directly from Monovar, hierarchical clustering is not necessarily the soundest way to make a tree from the Monovar calls. The use of hierarchical clustering ought to be better defended, or alternatively replaced with a more proper generic phylogeny tool (e.g., Paup* or MrBayes).

We would like to thank the reviewer for pointing out these issues. In order to compute a quantitative measure of the noise in Figure 4b, 4c, as well as 5b and 5c, we compute the difference in probabilities of their underlying matrices. To be precise, given the two dimensional probabilities matrices (M) of Figure 4 and 5, we computed the average difference of each matrix entry $M_{i,j}$ and its neighbours $M_{i-1,j-1}$, $M_{i+1,j-1}$, $M_{i-1,j+1}$, $M_{i+1,j+1}$. The results for SCIΦ in Figure 4 and 5 were 0.055 and 0.042 respectively. The values for Monovar were 0.216 for Figure 4 and 0.203 for Figure 5. Therefore, the qualitative measurement support our original statement that the SCIΦ results are less noisy compared to Monovar.

With regard to using a proper phylogeny tool for the tree inference, figures 4c and 5b are a graphical representation of the mutation probabilities of Monovar ordered using the inferred phylogenetic tree of SCIΦ. Because the tree inference scheme of SCIΦ is very similar to the one described in SCITE we do make use of a phylogenetic tree reconstruction tool tailored towards single-cell data. By contrast, Figure 4d and 5a illustrate the scenario where phylogenetic information is not used. For clustering, we used hierarchical clustering. However, as the reviewer pointed out, this method may not give the best possible phylogenetic tree. We edited the corresponding section in the manuscript to make that point more explicit. The second reason is that hierarchical clustering was also employed in the original publication of Monovar. In order to make the hierarchical clusterings more comparable we adopted the strategy of Monovar and assigned values of 0.5 to loci with 0 coverage. However, the results only changed marginally as can be seen in the updated figures 4d and 5a. For these two reasons we refrained from using Paup*, MrBayes, or the single cell tree reconstruction tools OncoNEM and SCITE.

5) The paper is well written, organized, and clear. Background and references are appropriate, although they could perhaps better cite work to date on phylogenetics from single-cell sequence data specifically. There was a recent review from Zafar et al. (Current Opinion in Systems Biology, 2017) that might be helpful there. I noted just one minor error: “substantial” should be “substantially” on p. 11 line 275.

We thank the reviewer for pointing out the article and cited it when describing the tree inference scheme (line 108: “Different approaches for single-cell phylogeny reconstruction have been proposed \citep{Zafar2018} ...”).

Reviewer #2 (Remarks to the Author):

The authors propose a brand new computational method to infer mutations for single cell sequencing data based on jointly inferring the phylogenetic tree of cells. The idea is excellent and is expected to provide new methodological insights into the detection problem of somatic mutations. The implementation of the method is also elegant, reaching superior performance compared to Monovar. However, before acceptance for publication, some concerns should be clarified.

1. The basic idea of the proposed method is jointly inferring the cellular phylogenetic tree and the mutations at almost all the loci. In general, these two tightly-linked questions are solved separately. Employing the linkage of these two questions to refine the inference of each other is creative. However, it seems that the authors did not sufficiently evaluate the performance of the joint inference. Particularly, the authors only evaluated the accuracy of the new method for mutation detection, without validating the reliability of the inferred phylogenetic tree(s). If the phylogenetic tree is not significantly improved, the refined mutation detection needs a more persuasive explanation.

We thank the review for his comment. The tree inference scheme employed by SCIΦ is based on the one of SCITE [1] that has been shown to accurately and reliably infer tumor phylogenies. To further substantiate this claim in Supplementary Section “Tree inference performance” we specifically analyzed the performance of SCIΦ with respect to the phylogeny reconstruction and show that it reliably finds phylogenetic trees very close to the true phylogenetic tree.

[1] Jahn, K., Kuipers, J., & Beerenwinkel, N. (2016). Tree inference for single-cell data. *Genome biology*, 17(1), 86.

2. According to my understanding, the joint inference is conducted separately for each locus. However, the true phylogenetic tree of a specific cell population seems to be unique, and more importantly the contribution of each locus to the construction of the cell phylogenetic tree varies greatly. My concern is how consistent the inferred trees were for different loci. If the consistency is under expectation, how did the trees guide the inference of mutations? The authors should show example trees inferred from different loci so that the readers can get intuitive impressions of the performance of the proposed method.

We appreciate the reviewer's comment and would like to point out that the joint inference is not conducted separately for each mutation, but all mutations are evaluated using the same phylogeny. This is described in Section 2.4 where we describe the likelihood of a specific tree realization with the mutation attachment parameter. It is more formally defined in equation 9. We changed the line 124 to "The first set of products describes the loci identified to be likely mutated (Section 2.2) which are placed on the tree and used together to infer its phylogenetic structure.", to show that the mutations are used together and not separately.

3. As the authors mentioned, the joint inference results in huge computational complexity. The authors should show the running time and memory usage for different datasets and compare to other softwares. Only Monovar is not sufficient to demonstrate the relative performance of the proposed method. Mutect, Strelka and other frequently used tools should also be included.

We thank the reviewer for his suggestion. In order to demonstrate that traditional bulk sequencing variant callers are not well suited for single cell variant calling we describe an additional benchmark in Supplementary Section "Comparison of single-cell and bulk sequencing variant callers". Here, we compare the performance of Monovar and SCIΦ to VarScan2, a widely used mutation caller. We show that the single-cell specific mutation callers, SCIΦ and Monovar, both outperform the bulk sequencing mutation caller. Mutect, Mutect2, and Strelka are usually used in a tumor-normal matched sample setting where a tumor and control bulk sample are directly compared, which is not available in our setting. In addition, we now provide run times and memory peaks for the benchmarked tools in Supplementary Section "Computational resources". In summary, Monovar as well as SCIΦ have low memory requirements and can be run within hours.

4. Only two real datasets were included to evaluate the tool performance, and the cell numbers were only 16 and 255. Datasets with more cells should be evaluated.

The reviewer raises an important point. We conducted two additional analyses on real data sets. The first is derived from an isogenic fibroblast cell line and described in Supplementary Section "Performance comparison using isogenic cell line data" (also see comment to reviewer 1 point 3). Here, 19 cells that harbored more than 50,000 mutations were analyzed. In accordance with the simulations, SCIΦ has a higher F1 score. It is worthwhile noticing that Monovar's performance is better with respect to precision. However, as we show in Supplementary Section "Performance

comparison using isogenic cell line data” there is strong evidence that many of SCIΦ’s false positive mutations are indeed true positives. Second, we analyzed 370 cells from a high-grade serous ovarian cancer patient with 43 mutations in Supplementary Section “Performance comparison on a high-grade serous ovarian cancers patient”. The result is very similar to the other real data sets with SCIΦ providing a much cleaner assignment of mutations to cells than Monovar. It is also interesting to observe that Monovar assigned two mutations to the rightmost cluster of cells in Supplementary Figure 16b that SCIΦ does not assign. A closer investigation of the corresponding loci revealed that almost all of these mutations have a variant allele frequency below 0.01. Hence, it appears more likely that these mutations are in fact only sequencing errors.

5. Another concern is the robustness of the inference. Due to the computational complexity, the authors used MCMC to complete the inference. The authors should demonstrate the robustness of the inferring results by adding replications. Furthermore, quality control is a critical step before mutation detection. The authors should also show whether the inference is robust to various quality control criteria.

We thank the reviewer for his comment. In fact, robustness is a very important criterion. In order to determine whether SCIΦ converged on a particular data set we compare the result of two independent runs which started with different seeds. In order to do so, we compute the correlation of the mutation to cell assignment. We did so for all real data sets. The correlation coefficient was always above 0.99, demonstrating the robustness of SCIΦ.

With respect to influence of different quality threshold we agree with the reviewer that they might have an influence. However, because there are many different quality criteria with different settings it is difficult to address them. In general, changing the quality cutoffs to be less restrictive will include more noise, which might influence the tree reconstruction. In Supplementary Section “Violation of the infinite sites assumption” we address the effects of systematic violations of the infinite sites assumption and show the robustness of SCIΦ. Therefore, we show that different noise levels only have a moderate impact on the performance of SCIΦ. Further, we also explicitly state how the results in the manuscript were obtained to provide an example of a possible analysis setting and its quality cutoffs.

Reviewer #3 (Remarks to the Author):

The manuscript by Singer et al describes SCI, a new statistical approach for detecting variants from single cell DNA sequence data. This is a very important topic as currently most single cell DNA analysis projects use algorithms that are designed for bulk samples and have poor performance. Their new statistical framework exploits the power gained by inferring the underlying phylogenetic tree of a collection of multiple cells from the same sample to mitigate errors in single cell data, including false positive errors, coverage unevenness, allelic dropout and loss of heterozygosity. Using the controlled experiment of a simulated single cell dataset, they demonstrate that SCI has greatly improved sensitivity and specificity over the currently existing single cell mutation identification methods and the widely used algorithms such as Samtools and GATK UnifiedGenotyper. They then apply their new method to real single cell DNA sequencing datasets from two human tumor samples (a

triple-negative breast cancer and an acute lymphoblastic leukemia cancer), detect somatic mutations and infer the clonal structures of the two tumors. The manuscript is very well written and interesting, although this reviewer has the following concerns:

1. The authors compared SCI with Monovar, another single cell SNV detection method and demonstrated that SCI outperforms Monovar. The performance improvement of Monovar mostly comes from the error model used by the algorithm. The authors of Monovar cultured an isogenic cell line, then sequenced multiple individual cells from the cell line. They then compared single cell data against bulking sequencing results of the same cell line. Assuming mutational differences are all errors, they built the errors model of single cell DNA sequencing and filtered these errors from real single cell datasets. Monovar also counts only mutations that are shared by at least two cells. So, my question here is, where does the performance improvement of SCI come from?

We thank the reviewer for his question. The performance improvement of SCIΦ is not only due to the error model, but mostly due to sharing information via the phylogenetic tree structure. This approach allows for identifying mutations much more reliably in the presence of drop-out events, because of the constraints imposed by the tree structure. An example is shown in the figure below. Here the dark green squares indicate the occurrence of a mutation and the grey boxes show the genotype of the cells below. The cells are depicted as leaves with nucleotide counts for each mutation. The numerator of the nucleotide counts represents the number of reads supporting the mutation while the denominator depicts the coverage. In the example, the second cell from the left shows zero out of 10 reads supporting the second mutation. However, because the neighbouring cells of the cluster show support for the mutation, SCIΦ is able to identify this observation as a drop-out event and assigns the mutation to the cell. As a result, SCIΦ shows a strong increase in sensitivity, i.e., it can pick up many more mutations without loss of precision.

2. Is SCI able to detect small insertions and deletions using single cell DNA data? Could the authors evaluate the performance of SCI for detecting INDELS?

We would like to thank the reviewer for this comment. At the moment SCIΦ does not support the identification of insertions or deletions. This task is complex and would require an additional model. Because of noise in the data, real world data sets often contain contradicting indel information for a specific locus on the genome. For

example, regions containing homopolymers often cause errors during the sequencing, resulting in some reads containing too many and some reads containing too few characters of the homopolymer character. However, this important question should be approached in future versions of SCIΦ.

3. The authors benchmarked SCI using a simulated single cell DNA sequencing dataset. Even though they tried to spike-in random false positive errors, allelic dropouts etc., the error pattern of simulated data is always different from that of real single cell data. Will the authors be able to benchmark SCI using an isogenic cell line data?

Yes, we added this additional benchmark as described in answers to comments 1.3 and 2.4 in Supplementary Section “Performance comparison using isogenic cell line data”. We show that similar to the simulations, SCIΦ performs favorably in terms of F1 score. In addition, we provide detailed numbers on false positives and false negatives and explain the difficulties of using such a data set as ground truth.

4. The authors applied SCI to two real single cell datasets. However not all the mutations from the two datasets are validated. My understanding is that many mutations, including mutations that are detected in just one cell, from the triple negative breast cancer tumor were validated using an alternative sequencing method. How does SCI perform using validated single cell mutations only?

We appreciate the reviewer’s comment. Since we do not have any biological material we cannot validate the mutations identified by SCIΦ and do not have a ground truth available. Therefore, as pointed out by the reviewer in his comment 3, we performed an additional benchmark on an isogenic cell line. For isogenic cell line data set we analyze in detail how many mutations are falsely identified and how many are missed. For the triple negative breast cancer patient, such data is not available. Nevertheless, as the reviewer suggested, we re-ran SCIΦ and restricted it to the set of validated mutations as reported in original study [1]. Because the authors of [1] only reported nonsynonymous clonal and subclonal mutation calls, the set of mutations was reduced to 443. However, the result on this reduced data set is very similar to the result using all mutations. One difference is that h2 moves closer to a1, a6, and a4 rather than to h4, h6, and h7. As can be seen in the figure below, the difference between h2 and h4, h6, and h7 is only 3 mutations that are probably drop-out events, but not identified as such because of the limited amount of mutations. In addition, h3 is not assigned to h5, h8, and a8, but rather to cluster also including a1, a4, a6, h2, h4, h6, and h7. Again, the separation is only very weak with h3 missing a single mutation in comparison to the remaining cells of the cluster. Further, in the tree of the complete data (Figure 4a of the manuscript) h3 is closer to a8, h5, and h8 on the basis of very few mutations. Therefore, the results are very similar and in general using more data is preferable. Also note that the authors of [1] make the point that even most of the mutations which could not be validated are likely real biological mutations.

Figure 1: Overview graph of the TNBC using only validated mutations.

[1] Wang, Y., Waters, J., Leung, M. L., Unruh, A., Roh, W., Shi, X., ... & Multani, A. (2014). Clonal evolution in breast cancer revealed by single nucleus genome sequencing. *Nature*, 512(7513), 155.

5. In figure 4a, the authors inferred the clonal structures of the triple negative breast cancer using mutations detected by SCI and classified cells h3, h5, h8 and a8 into one sub-clone. However, the Nature paper by Wang et al. suggests that cells h3, h5 and h8 are hypodiploidy cells while a8 is an aneuploidy cell. These two groups of cells have very different genotypes based on flow sorting results and single cell mutations. How were they grouped together?

Indeed, there is a difference between the original publication and the results of SCIΦ. Especially a8 is assigned very differently. In the original publication (<https://www.nature.com/articles/nature13600/figures/3>, Figure 3), a8 shares several mutations exclusively with a2, a3, and a7. This is not the case for our results. Unfortunately there is not direct assignment of mutations to cells in the supplements of the original publication. Therefore, a direct comparison is impossible. However, there are two mutations that are explicitly shown to be shared in the cluster of a2, a3, a7 and a8 in Figure 3 of the original paper [1]. TGFB2 is mutated in three cells

according to the supplements, but only mutated in two cells in Figure 3. In addition, CHRM5 is mutated in 2 cells according to the supplements, but from Figure 3, CHRM5 seems to be mutated in 4 cells. Further, in our pileup files we checked the support of a8 for the two mutations, which was zero in both cases. Therefore, we cannot explain how these mutations were assigned to a8 in the original publication.

[1] Wang, Y., Waters, J., Leung, M. L., Unruh, A., Roh, W., Shi, X., ... & Multani, A. (2014). Clonal evolution in breast cancer revealed by single nucleus genome sequencing. *Nature*, 512(7513), 155.

6. SCI counts only mutations that are shared by at least two cells. However, when the authors studied the clonal structures of the triple negative breast cancer using mutations identified by SCI, they found that cell h1 is an orphan cell. H1 shares all its mutations with at least one other cells but it is also an orphan. How come?

H1 indeed shares all mutations with the other cells, and without any other mutations it is placed directly under the root, which harbors the mutations in common between all cells.

7. In figure 4d, the authors classified cells from the triple negative breast cancer using Monovar mutations. However, the classification result is very different from the result of the Monovar paper. What has changed?

We thank the reviewer for this comment. The results are not that different in the sense that most of the clusters contain the same cells. The remaining difference can likely be explained by the heavy post filtering of mutations in the Monovar paper. Further, instead of using the probabilities assigned to the mutations the authors of Monovar simply used a binary assignment of mutations to nodes. Either a mutation was present in a given cell or not. In addition, all loci with coverage below 6 were assigned a value of 0.5 in the original publication. Removing the full information of the probabilities of the mutation assignments as computed by Monovar will affect their plots. All of these reasons likely contribute to the differences between clusterings.

REVIEWERS' COMMENTS:

Reviewer #1 (Remarks to the Author):

I believe as in my review of the first submission that this is a useful and timely paper on an important problem for which new tools are much needed. I am satisfied with the responses of the authors to my prior critiques. The authors have clarified several prior points of confusion for me. Their new experiments show the method to be robust to several potential sources of error, such as violations of the infinite sites assumption or repeat structure of real genomes. Where the authors have declined to implement requested revisions, I believe they have given reasonable justifications. While the method does have some limitations, e.g., not handling CNVs, these are reasonably considered for future work.

I have no other substantive critiques to raise. I noted only one small error introduced in the revisions: on the bottom of p. 5 of the supplement, the phrase "case were only" should read "case where only".

Reviewer #2 (Remarks to the Author):

The authors have addressed all my concerns and the manuscript is acceptable in my opinion.

Reviewer #3 (Remarks to the Author):

The Authors have addressed my concerns. I have no further comments.